# Durability of Gadolinium Zirconate/YSZ Double-Layered Thermal Barrier Coatings under Different Thermal Cyclic Test Conditions

**DOI:** 10.3390/ma12142238

**Published:** 2019-07-11

**Authors:** Satyapal Mahade, Nicholas Curry, Stefan Björklund, Nicolaie Markocsan, Shrikant Joshi

**Affiliations:** 1Department of Engineering Science, University West, 46186 Trollhattan, Sweden; 2Treibacher Industrie AG, Auer von Welsbachstr, 1, A-9330 Althofen, Austria

**Keywords:** double-layered TBC, gadolinium zirconate, suspension plasma spray, thermal cyclic fatigue, burner rig test, yttria stabilized zirconia

## Abstract

Higher durability in thermal barrier coatings (TBCs) is constantly sought to enhance the service life of gas turbine engine components such as blades and vanes. In this study, three double layered gadolinium zirconate (GZ)-on-yttria stabilized zirconia (YSZ) TBC variants with varying individual layer thickness but identical total thickness produced by suspension plasma spray (SPS) process were evaluated. The objective was to investigate the role of YSZ layer thickness on the durability of GZ/YSZ double-layered TBCs under different thermal cyclic test conditions i.e., thermal cyclic fatigue (TCF) at 1100 °C and a burner rig test (BRT) at a surface temperature of 1400 °C, respectively. Microstructural characterization was performed using SEM (Scanning Electron Microscopy) and porosity content was measured using image analysis technique. Results reveal that the durability of double-layered TBCs decreased with YSZ thickness under both TCF and BRT test conditions. The TBCs were analyzed by SEM to investigate microstructural evolution as well as failure modes during TCF and BRT test conditions. It was observed that the failure modes varied with test conditions, with all the three double-layered TBC variants showing failure in the TGO (thermally grown oxide) during the TCF test and in the ceramic GZ top coat close to the GZ/YSZ interface during BRT. Furthermore, porosity analysis of the as-sprayed and TCF failed TBCs revealed differences in sintering behavior for GZ and YSZ. The findings from this work provide new insights into the mechanisms responsible for failure of SPS processed double-layered TBCs under different thermal cyclic test conditions.

## 1. Introduction

Thermal barrier coatings (TBCs) enhance the efficiency of a gas turbine engine by allowing them to operate at higher temperatures, in order to lower engine emissions and improve fuel economy [1]. At higher operating temperatures (>1200 °C), the existing state-of-the-art top coat TBC candidate, 7–8 wt. % yttria stabilized zirconia (YSZ), has several limitations such as phase instability, high sintering rates, etc. [2,3]. Another major drawback of YSZ as a top coat material is also its susceptibility to CMAS infiltration above 1200 °C, which limits TBC longevity [4,5,6,7].

Alternative ceramic top coat materials for TBC application such as gadolinium zirconate (GZ), have been shown to possess lower thermal conductivity and excellent phase stability compared to YSZ at high temperatures [8]. The other advantage of GZ over YSZ is its excellent CMAS (Calcium- Magnesium-Alumino-Silicates) attack resistance [9]. However, GZ has drawbacks such as inferior fracture toughness and poor thermochemical compatibility with alumina (thermally grown oxide), which limits its durability [10,11]. To overcome these drawbacks, double-layered TBCs with GZ as the top layer and YSZ as the base layer have been used [12,13]. A double-layered, GZ/YSZ TBC design exploits the merits of YSZ (high fracture toughness close to the TGO/bond coat) and GZ (excellent CMAS resistance, low thermal conductivity etc.). Furthermore, the GZ/YSZ multi-layered TBCs were shown to be more durable compared to a YSZ single layer TBC [12]. Failure analysis of the GZ/YSZ multi-layered TBCs investigated so far with a layer thickness ratio of 4:1 for GZ and YSZ revealed spallation of the GZ layer close to its interface with YSZ and at a distance of less than 100 µm from the bond coat [12,14]. The reason for this was attributed to lower fracture toughness of GZ than YSZ [15]. Fracture toughness plays a crucial role in governing spallation in the ceramic as it resists crack propagation due to accumulation of stresses in the coating from various sources (coefficient of thermal expansion [CTE] mismatch between ceramic and metallic substrate, oxidation of bond coat, etc.) during thermal cycling [8]. The specific purpose of this study was to assess if the durability of GZ/YSZ double-layered TBC could be improved by moving the inferior fracture toughness material, i.e., GZ, further away from the bond coat by increasing the underlying YSZ layer thickness. By doing so, the relatively higher fracture toughness material (YSZ) is presented at the probable failure location, which could lead to improved durability by delaying the onset of failure. Durability of TBCs and their associated failure mechanisms vary with the test conditions employed (such as cycling conditions, exposure temperature and time etc.) [16]. Furthermore, the choice of substrate (composition) has an influence on the durability as it leads to differential CTE mismatch with the top coat, resulting in differing stress state in the TBC [17,18]. Previously, burner rig test comprising of short exposure time (75 s of heating cycle) and relatively lower exposure temperature (1350 °C), were employed to evaluate the performance of GZ/YSZ double-layered TBCs deposited on Hastelloy-X substrates [19].

In this work, three double-layered GZ/YSZ TBCs with variable YSZ thickness i.e., 400GZ/100YSZ, 250GZ/250YSZ and 100GZ/400YSZ, where the prefixed numbers represent layer thickness in µm, were deposited by SPS process. SPS process was opted over other TBC deposition techniques such as EB-PVD (Electron Beam- Physical Vapor Deposition) or APS (Atmospheric Plasma Spray) due to its capability to produce columnar microstructured TBCs via plasma route. Columnar microstructured TBCs are desirable for higher cyclic lifetime and higher erosion resistance [20,21]. The GZ/YSZ TBCs were subjected to two different thermal cyclic test conditions, one without a thermal gradient (thermal cyclic fatigue or TCF) and the other with a thermal gradient (burner rig test or BRT) across the TBC test specimen. For BRT, TBCs were deposited on Inconel-738 substrates and exposed to a relatively higher surface temperature and exposure time than the ones reported elsewhere [19]. The BRT and TCF failed TBC specimens were analyzed by SEM to gain insights into the mechanisms responsible for failure.

## 2. Experimental Details

Two different substrates, namely Hastelloy-X and Inconel-738, with dimensions 50 mm × 30 mm × 6 mm and 30 mm dia × 3 mm thickness were used for the TCF and BRT tests, respectively. The substrates were grit blasted to create a surface roughness of approximately 3 µm Ra. A NiCoCrAlY bond coat (AMDRY 386, Oerlikon Metco, Westbury, New York, NY, USA) was first deposited on the surface using High Velocity Air Fuel (HVAF) process (M3 gun, UniqueCoat, Oilville, Virginia, VA, USA). The thickness of bond coat was kept at 190 µm ± 10 µm in all the investigated specimens. HVAF sprayed bond coat may contain unmelted droplets and loosely bound particles on the surface which could hinder mechanical adhesion of the TBC. Therefore, grit blasting of the bond coated surface was carried out to get rid of the loosely bound particles and create a surface roughness of approximately 5 µm Ra. After grit blasting, the surface was cleaned with pressurized air to remove loosely bound grit particles.

Ethanol based, commercial 8YSZ and GZ suspensions supplied by Treibacher Industrie AG, Althofen, Austria were used. Both the 8YSZ suspension (AuerCoat YSZ) and the GZ suspension (AuerCoat Gd-Zr) comprised of powders with a median size of 500 nm and a solid load content of 25 wt. %.

The bond coated substrates were preheated prior to top coat deposition in order to remove volatile impurities from the surface. An axial-feed capable plasma torch (Axial III, Mettech, Vancouver, BC, Canada) was used to deposit the ceramic layers. Identical spray parameters (Table 1) were used for deposition of YSZ and GZ layers. The first double layered TBC variation had 400 µm thick GZ top layer and 100 µm thick YSZ base layer, which is denoted as 400GZ/100YSZ. Similarly, the second and third double-layered TBC variations are denoted as 250GZ/250YSZ and 100GZ/400YSZ.

Microstructure of the as-sprayed TBCs was characterized using a scanning electron microscope (HITACHI TM 3000, Hitachi High-Technologies Corporation, Tokyo, Japan). The porosity content in GZ and YSZ layers of as-sprayed and TCF failed TBCs were analyzed using an open software called ‘ImageJ’ (version 1.52p, University of Wisconsin, Wisconsin, US) [22] by considering twenty five different cross-sectional SEM micrographs at high magnification (5000×). Furthermore, column gaps in the TBC were not considered during porosity measurement. For simplicity, one TBC variation (250GZ/250YSZ) was chosen for the porosity measurement of GZ and YSZ layers in the as-sprayed condition since all the investigated TBCs were deposited using identical spray parameters. After TCF test, porosity evolution in GZ and YSZ layers for all three TBC variations was measured and compared with as-sprayed condition.

In the TCF test, the TBC specimens were exposed to 1100 °C for 1 hr and were later, cooled to 100 °C in 10 min using compressed air. This cycle was repeated until specimen failure, which was deemed to be 20% visible TBC spallation. After the completion of each cycle (heating and cooling), photograph of the specimens were taken by integrating a high-resolution camera with LabVIEW software in order to monitor the progress of failure. The temperature during the heating cycle was monitored using two thermocouples, which were placed at different locations in the heating chamber. Further details regarding the TCF test setup are disclosed in our previous work [23]. It should be noted that the TCF test was conducted in the absence of a thermal gradient across the TBC specimen. Three specimens of each coating variation were subjected to a TCF test and the mean value and standard deviation are reported.

In the burner rig tests (BRT), performed at Forschungszentrum Jülich, Germany, the TBC surface was exposed to a surface temperature of 1400 °C with the rear of the specimen maintained at a temperature of 1050 °C. The test specimen was heated periodically using natural gas/oxygen burners whereas the backside of the specimen was cooled using compressed air. The surface temperature was measured using a land infrared pyrometer whereas the backside (substrate) temperature was measured using a NiCr/Ni thermocouple. Each cycle involved 5 min. heating followed by 2 min of cooling. Failure criteria was 30% visual spallation. Further details regarding the test setup and TBC specimen geometry are disclosed elsewhere [24]. Two specimens were tested for each TBC variant and their mean values are reported.

## 3. Results and Discussion

### 3.1. Microstructure

The cross sectional SEM micrograph of the 400GZ/100YSZ double-layered TBC showed columnar microstructure, according to Figure 1a. A similar columnar microstructure of the GZ/YSZ double layered TBC processed by SPS process was reported elsewhere [25]. The thickness of top GZ layer was measured to be approximately 410 µm and the base YSZ layer was approximately 95 µm. The GZ/YSZ interface in this case seems to be continuous and free from delamination cracks, according to Figure 1b. This is highly desirable since any cracking or discontinuity at the interface can promote delamination and adversely influence durability. Similarly, the cross sectional SEM micrographs of 250GZ/250YSZ and 100GZ/400YSZ TBCs showed a columnar microstructure along with a delamination free GZ/YSZ interface, according to Figure 1c–f, respectively.

### 3.2. Porosity

Porosity content in the individual layers comprising the TBC specimen, i.e., GZ and YSZ, were measured in as-sprayed condition for 250GZ/250YSZ. The YSZ layer showed a higher porosity content than the GZ layer, although the spray parameters and suspension properties were kept the same, see Table 1 and Figure 2. It should be noted that GZ has a lower melting temperature (2570 °C) than YSZ (2700 °C). Therefore, the GZ splats undergo a greater degree of melting than YSZ. According to the understood theory of plasma spraying, this should result in a relatively denser coating for GZ than for YSZ. This explains the higher porosity content observed for YSZ than for GZ. Similar findings of YSZ showing higher porosity compared to GZ were reported in the past under similar processing conditions [25].

The porosity content in a TBC can be expected to progressively change with time when subjected to a burner rig test as a result of sintering. Moreover, the porosity can plausibly also vary along the TBC thickness due to the presence of a thermal gradient across the test specimen, leading to greater sintering in the ceramic layer closer to the burner. Therefore, it is a challenge to compare the porosity content in a TBC before and after exposure to BRT. However, in a TCF test, the lack of a thermal gradient across the test specimen ensures that the extent of sintering in the ceramic material is independent of thickness across the TBC. Therefore, in this work, porosity evolution in GZ and YSZ layers were compared in as-sprayed and TCF failed condition for 400GZ/100YSZ, 250GZ/250YSZ and 100GZ/400YSZ TBCs. After the TCF test, the individual GZ and YSZ layers showed a reduction in porosity compared to the as-sprayed condition, according to Figure 2. Sintering is known to lead to an increase in stiffness of the TBC, which could result in loss of strain tolerance and eventually lead to TBC failure [26,27]. Therefore, it is desirable to have a ceramic top coat material which can provide better sintering resistance. Furthermore, according to the mean porosity values in Figure 2, the YSZ layer showed higher reduction in porosity than GZ in all the three variants of TCF tested TBCs when compared to the as-sprayed condition. However, the error bar consideration shows no significant difference in the sintering resistance of GZ and YSZ. Cao et al. reported higher sintering resistance of rare earth based pyrochlores than YSZ [28]. The reason for higher sintering resistance of pyrochlores than YSZ was attributed to the fact that the oxygen anion vacancies in pyrochlores are arranged in an orderly fashion compared to YSZ [28].

### 3.3. TBCs Subjected to TCF Test 

In TCF, the test conditions are relatively harsh due to the fact that the top coat, bond coat and the substrate are exposed to the same temperature, i.e., there is no thermal gradient across the TBC test specimen. The TCF test results indicate that the GZ/YSZ double-layered TBC with higher YSZ thickness (100GZ/400YSZ) showed lower durability whereas the TBC with lower YSZ thickness (400GZ/100YSZ) showed higher durability, as depicted in Figure 3. In a TBC, the bond coated surface could be accessed (oxidized) by oxygen via two possible routes i.e., from the open porosity through TBC and the oxygen ion vacancies in the crystal structure of the ceramic. It is speculated that the oxygen penetration resistance of rare earth zirconate based pyrochlores is higher than YSZ due to their cubic crystal structure having a systematic arrangement of the oxygen ion vacancies [28,29]. Moreover, the porosity content in the GZ layer has also been shown to be lower than that in the YSZ layer, thereby further contributing to the better oxygen penetration resistance.

The cross sectional SEM micrograph of failed 400GZ/100YSZ showed column gap widening in the double-layered TBC, according to Figure 4a. The reason for column gap widening could be attributed to the tensile stresses in the TBC during the heating cycle, where the metallic substrate expands more than the ceramic. The photograph of failed TBC specimen showed TBC spallation and the oxidized bond coat surface being exposed, according to Figure 4b. At high magnification, the failure in the TGO layer due to horizontal crack propagation could be seen, according to Figure 4c. Similar failure mode in the GZ/YSZ double layered TBCs when subjected to TCF test was reported elsewhere [12]. The thermally grown oxide (TGO) layer at failure showed a thickness of approximately 6–7 µm. Lu et al. and Smialek et al. previously reported the critical TGO thickness at failure in TCF tested specimens to be approximately 7–8 µm [30,31,32]. The failed specimen does not show blue failure, indicating the presence of alumina in the failed region of TGO [33].

The cross sectional SEM micrograph of the failed 250GZ/250YSZ and 100GZ/400YSZ double layered TBCs also showed failure in the TGO layer, according to Figure 5a and Figure 6a. The TGO thickness at failure was also measured to be approximately 6–7 µm in these TBC variations. The photographs of the TBC after failure also showed spallation of the ceramic coating from the test specimen edges, according to Figure 5b and Figure 6b. The high magnification cross sectional SEM micrograph of failed TGO layer in the case of 100GZ/400YSZ is shown in Figure 6c.

Failure in the double-layered TBC variations (400GZ/100YSZ, 250GZ/250YSZ, 100GZ/400YSZ) when subjected to TCF test appears to be similar, i.e., spallation of TBC due to failure in the TGO. However, their TCF durability differs, with the 400GZ/100YSZ showing the highest durability and 100GZ/400YSZ showing the lowest. Oxidation of bond coat and attaining critical TGO thickness has been previously reported to be the limiting step governing the durability of TBCs under a TCF test, and this has been reaffirmed by the present results.

In the TCF test, the relatively longer exposure to high temperature allows sufficient time for oxidation of the bond coat, while the ceramic coating simultaneously undergoes a degree of sintering. Furthermore, the CTE mismatch between the metallic substrate and ceramic coating during the heating and cooling cycle leads to accumulation of strain energy in the coating. These three mechanisms (oxidation of bond coat, sintering, CTE mismatch) compete for the TBC failure and reaching a critical TGO thickness via bond coat oxidation appears to be the dominant failure mode under TCF test conditions.

### 3.4. TBCs Subjected to BRT

The TBC surface temperature in BRT test was chosen as 1400 °C in order to replicate the desired service temperature of an aero engine. In BRT, the durability of double-layered TBCs showed a ranking similar to that during TCF testing, where the TBC with higher YSZ thickness (100GZ/400YSZ) in the GZ/YSZ double-layer coating showed lower durability and vice-versa, although the absolute values of the lifetime were different, see Figure 7. Previous findings demonstrated that the absolute values of durability results would improve (longer lifetime) when the TBC surface temperature is lowered [23]. Furthermore, ranking of the coatings, in terms of durability, were shown to be the same with lower exposure temperature [23].

Failure analysis of the BRT 400GZ/100YSZ TBC showed spallation of top GZ layer from region close to interface with YSZ, according to Figure 8a,c. Exposure of TBCs to thermal cyclic test results in accumulation of strain energy in the ceramic coating. This could be attributed to the coefficient of thermal expansion (CTE) mismatch between the ceramic coating (10.4 × 10^−6^/K for GZ and 11.5 × 10^−6^/K at 30−1000 °C [8]) and the metallic substrate (16–17 × 10^−6^/K at 1000 °C [34]) during the heating cycle (resulting in tensile stresses in the TBC) and cooling cycle (compressive stresses in the TBC). Viswanathan et al. used the concept of available elastic energy to explain the failure modes observed in their findings for APS processed GZ/YSZ TBCs subjected to thermal cyclic test [35]. In their findings, it was reported that TBC failure would occur when the available elastic energy exceeds the critical stress intensity factor (fracture toughness) of the ceramic [35]. YSZ has higher fracture toughness than GZ (approximately double) [15]. Recently, Zhou et al. reported the fracture toughness of SPS processed, porous, columnar microstructured YSZ TBC was approximately 1.0 Mpa.m^1/2^ [36] whereas, for SPS processed porous GZ, fracture toughness was 0.48 Mpa.m^1/2^ [10]. Therefore, in a GZ/YSZ double-layered TBC, it is expected that the GZ layer would allow crack propagation with relative ease compared to YSZ. After reaching a certain number of cycles in the burner rig, the stored elastic strain energy in the GZ/YSZ system presumably exceeds the fracture toughness of GZ and, hence, results in spallation of the GZ layer from a region close to GZ/YSZ interface. Similar horizontal cracks close to GZ/YSZ interface were reported previously elsewhere for GZ/YSZ double-layered TBCs processed by APS [12]. The photograph of failed 400GZ/100YSZ TBC showed failure in the ceramic layer, leaving behind some part of the intact ceramic layer, see Figure 8b. Furthermore, the blue failure appearance in the photograph suggests the presence of NiO and other spinels. The TGO layer thickness at failure, as shown in Figure 8d, was measured to be approximately 2 µm, which happens to be lower than the critical TGO thickness (as seen in TCF tested specimens).

In the case of failed 250GZ/250YSZ TBC, horizontal crack parallel to GZ/YSZ interface was observed. Furthermore, the location of horizontal crack shifted away (approximately 250 µm) from the bond coat compared to the 400GZ/100YSZ failed TBC, see Figure 8a and Figure 9a. The reason for such a shift in failure location could be due to the presence of higher fracture toughness material, i.e., YSZ, at the previously reported failure location. However, the durability results did not show improvement over 400GZ/100YSZ. The photograph of failed 250GZ/250YSZ also confirmed the spallation in the ceramic layer, see Figure 9b. High magnification SEM micrograph of the GZ/YSZ interface showed interlinking of horizontal and vertical cracks in GZ layer, according to Figure 9c, which led to spallation of GZ layer. TGO thickness at failure in this case was measured to be less than 2 µm. Furthermore, the YSZ layer close to TGO was free from cracks, according to Figure 9d.

In the case of failed 100GZ/400YSZ TBC, the cross sectional SEM micrograph shows delamination of the top GZ layer along with a thin remnant layer of GZ, according to Figure 10a. The failure location shifted further away from the bond coat (approximately 400 µm from the bond coat). The BRT lifetime was determined based on the photographs captured after each cycle. In this case, the test specimen was exposed to BRT conditions for longer cycles than its lifetime. Therefore, after the spallation of the GZ layer, horizontal cracks in YSZ appeared at two different locations; one close to the free surface and one close to the bond coat, see Figure 10c,d. The reason for horizontal crack propagation close to the bond coat could be attributed to the mismatch in coefficient of thermal expansion (CTE) between YSZ and metallic substrate. When the stored elastic energy in the TBC due to CTE mismatch exceeds the fracture toughness of the material, a crack propagates through the coating, leading to spallation. Previous findings on failure of the YSZ single layer TBC when subjected to a burner rig test reported similar horizontal crack propagation in the YSZ layer close to the bond coat [14]. The photograph also suggests that the failure occurred in the ceramic layer, as some part of the ceramic was still intact after the test, according to Figure 10b.

In the BRT, the relatively shorter time of exposure to high temperature during each cycle prevents sufficient time for oxidation of the bond coat and hence TGO growth is restricted, as seen in this work (<2 µm at failure in all the tested coatings). Simultaneously, the ceramic coating also undergoes sintering to some extent at a relatively higher exposure temperature, with the sintering rate being higher close to the surface than near the bond coat. The evidence of sintering was reported in our previous work for SPS-processed GZ and YSZ TBCs when subjected to BRT (same test rig) even at lower surface temperature (1300 °C) and time, see [14]. If the TBC failure occurs near the surface, it could be argued that the potential cause for failure was due to sintering. The failure analysis of BRT specimens in this study did not reveal failure at the TBC surface. Furthermore, the failure was not observed in the TGO (<2 µm thickness) layer under BRT conditions as the TGO thickness at failure was well below the critical TGO thickness On the other hand, the CTE mismatch between ceramic coating and the metallic substrate during BRT results in accumulation of strain energy in the TBC. When the strain energy in the coating exceeds the fracture toughness of the material, failure occurs. In the current work, all the three double-layered TBC variations failed in GZ (inferior fracture toughness material) layer close to interface with base YSZ layer, indicating CTE mismatch as the potential cause for failure. It seems that the three mechanisms (oxidation of bond coat, sintering and CTE mismatch) compete with each other for TBC failure during BRT and, in this case, the CTE mismatch between the ceramic and metallic substrate wins the race. Based on the author’s hypothesis, with an increase in YSZ layer thickness in the GZ/YSZ double-layered TBC, the BRT lifetime should have been higher. However, the BRT results obtained in this work contradict the author’s hypothesis, although the failure mechanisms were similar for all the three GZ/YSZ double-layered TBC variants. It is worthy to mention that GZ has lower thermal conductivity than YSZ (approximately 30% lower) [8,23]. Improved thermal insulation in 400GZ/100YSZ TBC due to lower YSZ thickness leads to lower bond coat temperature, which results in lower CTE mismatch and stress levels in 400GZ/100YSZ than 100GZ/400YSZ TBC. This could be one possible explanation for the improved durability of 400GZ/100YSZ than 100GZ/400YSZ. BRT results indicate that, in addition to the TBC fracture toughness, thermal insulation property of the TBC also plays an important role in governing durability.

## 4. Conclusions

In this work, gadolinium zirconate/YSZ double-layered TBCs with varying GZ/YSZ thickness combinations were investigated to evaluate the hypothesis that an increase in the YSZ layer thickness would enhance TBC durability. It was conjectured that this would provide a bigger region of higher toughness in the immediate vicinity of the bond coat to consequently delay cracking in the tougher YSZ layer or shift the probable failure-prone GZ region further away from the bond coat. The as-sprayed TBCs were subjected to different thermal cyclic test conditions, i.e., BRT (with temperature gradient) and TCF (without temperature gradient). Test results obtained in this work oppose the author’s hypothesis, as an increase in YSZ layer thickness in the GZ/YSZ double-layered TBC led to inferior durability under BRT and TCF test conditions. A possible explanation for inferior durability of 100GZ/400YSZ under BRT could be due to its higher thermal conductivity than 400GZ/100GZ (GZ has 30% lower thermal conductivity than YSZ), resulting in severe stress state (due to higher CTE mismatch) in the 100GZ/400YSZ coating. On the other hand, in the case of TCF, inferior durability of TBC with higher YSZ thickness (100GZ/400YSZ) could be due to the higher oxygen penetration resistance of GZ than YSZ.

Failure modes under TCF and BRT conditions were found to differ in the investigated TBCs. Among the three possible mechanisms for TBC failure i.e., sintering of the ceramic, oxidation of bond coat and CTE mismatch between the top coat and bond coat; oxidation of the bond coat and reaching a critical TGO thickness were found to be the reasons for TBC failure under TCF test conditions. In contrast, in the case of BRT, it was shown that CTE mismatch between the ceramic coating and metallic substrate dictates the TBC failure. Furthermore, with an increase in the YSZ layer thickness in the GZ/YSZ double-layered TBC, failure location shifted northwards from the bond coat, but remained in the GZ layer close to the interface with YSZ. In this work, it was also shown that an increase in fracture toughness at the probable failure location does not necessarily improve the durability. Other factors (such as thermal conductivity of the top coat) could play an important role in dictating the durability of the TBC.

Although the failure modes under different thermal cyclic conditions differed for the investigated TBCs, durability was shown to be superior for 400GZ/100YSZ TBC. Further improvement in the durability of 400GZ/100YSZ TBC could be achieved by opting for a denser GZ microstructure (due to improved fracture toughness) as failure in BRT in this work was shown to be in the GZ layer close to the YSZ interface.

## Figures and Tables

**Figure 1 materials-12-02238-f001:**
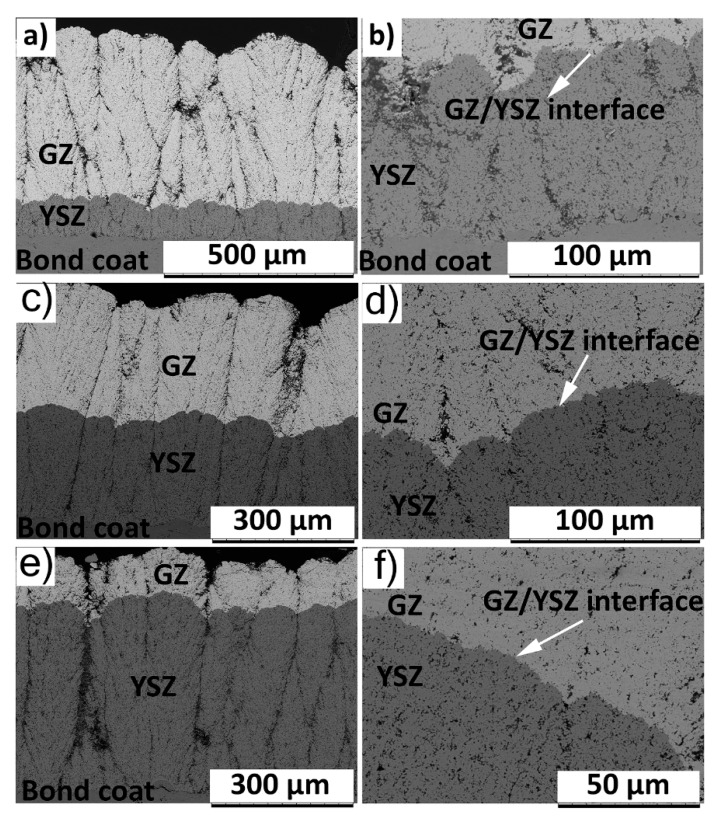
SEM (scanning electron microscope) micrograph of as-sprayed TBCs: (**a**) 400GZ/100YSZ cross section; (**b**) GZ/YSZ interface of 400GZ/100YSZ; (**c**) 250GZ/250YSZ cross section; (**d**) GZ/YSZ interface of 250GZ/250YSZ; (**e**) 100GZ/400YSZ cross section; (**f**) GZ/YSZ interface of 100GZ/400YSZ.

**Figure 2 materials-12-02238-f002:**
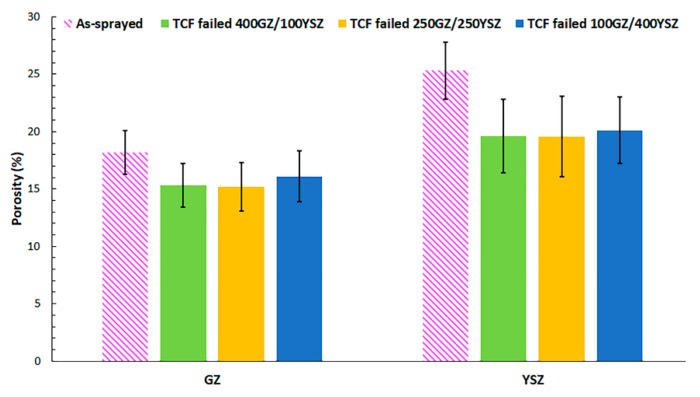
Porosity content of as-sprayed GZ and YSZ layers and after failure.

**Figure 3 materials-12-02238-f003:**
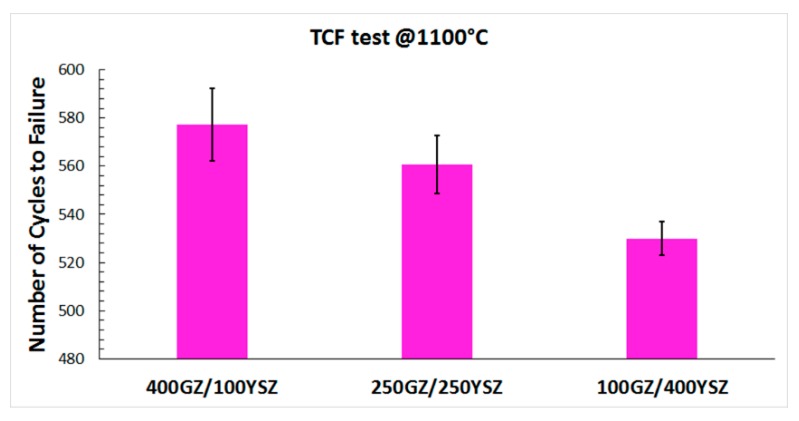
Thermal cyclic fatigue (TCF) life of GZ/YSZ double-layered TBCs.

**Figure 4 materials-12-02238-f004:**
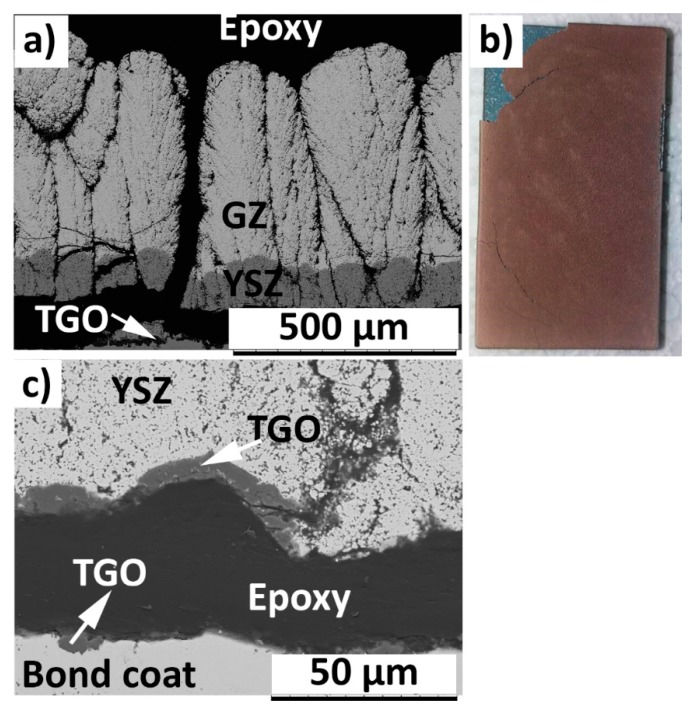
SEM micrograph of TCF (thermal cyclic fatigue) failed (570 cycles) 400GZ/100YSZ (**a**) Cross section and (**b**) Photograph (**c**) TGO (thermally grown oxide).

**Figure 5 materials-12-02238-f005:**
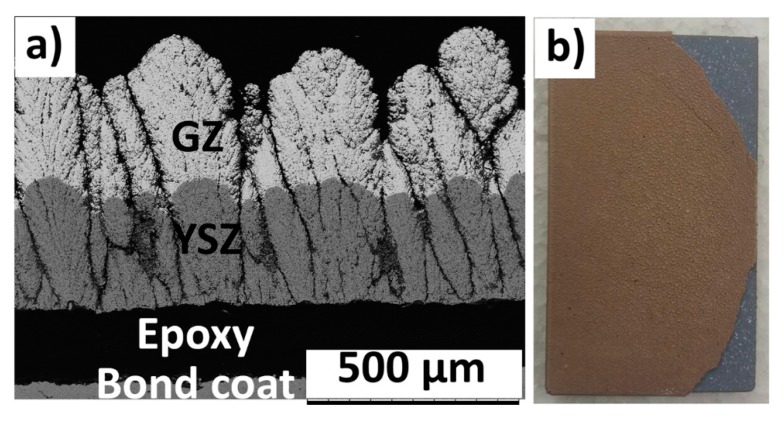
SEM micrograph of TCF failed (552 cycles) 250GZ/250YSZ (**a**) Cross section and (**b**) Photograph.

**Figure 6 materials-12-02238-f006:**
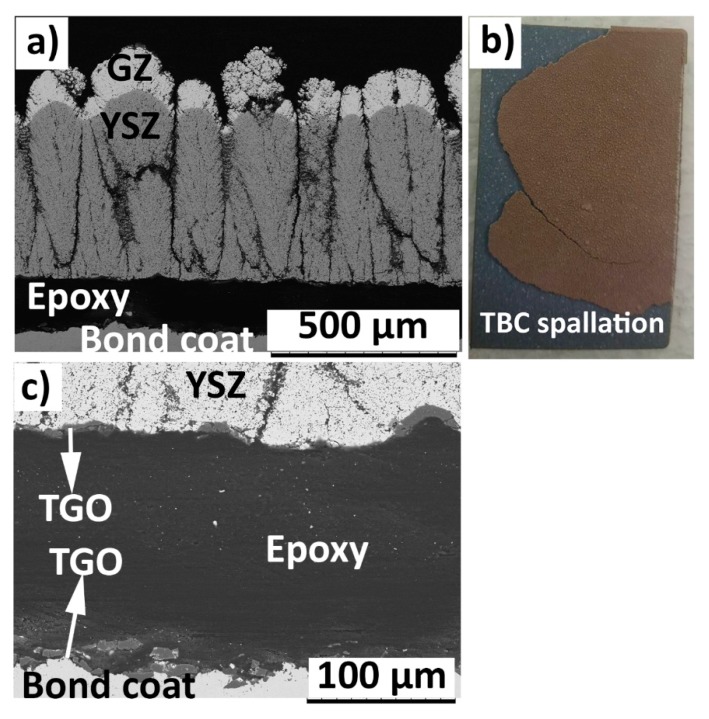
SEM micrograph of TCF failed (528 cycles) 100GZ/400YSZ (**a**) Cross section; (**b**) photograph and (**c**) TGO.

**Figure 7 materials-12-02238-f007:**
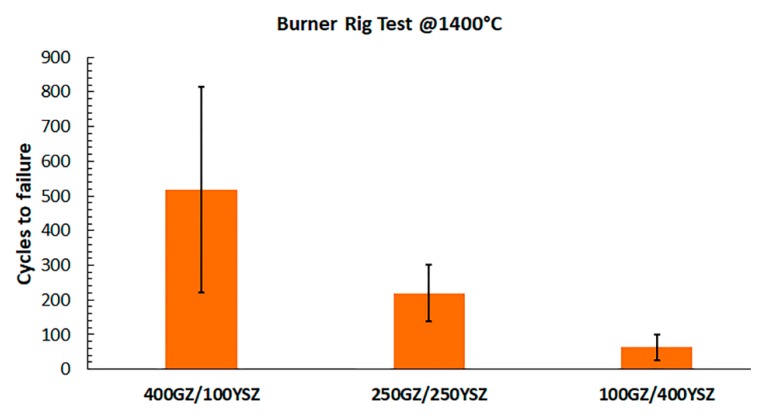
Burner rig test (BRT) life of GZ/YSZ double-layered TBCs.

**Figure 8 materials-12-02238-f008:**
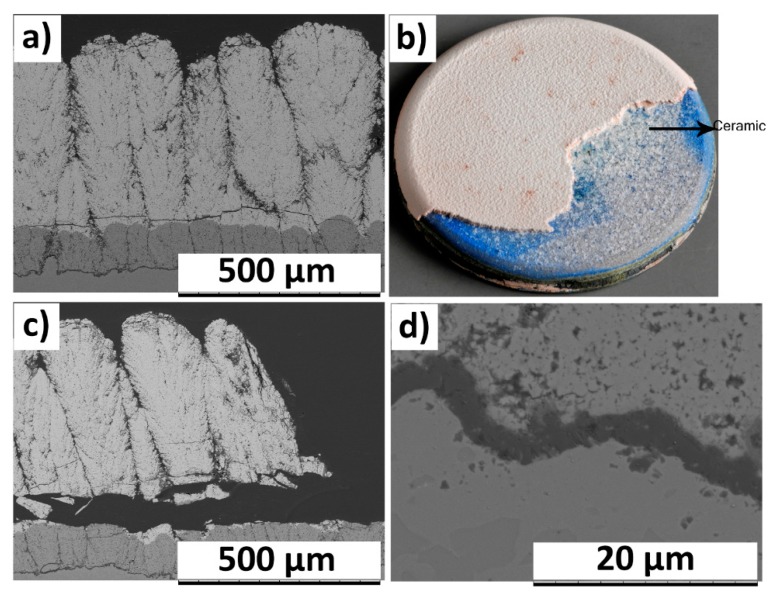
SEM micrograph of BRT (burner rig test) failed (814 cycles) 400GZ/100YSZ (**a**) Cross section; (**b**) photograph; (**c**) cross sectional view at a different location; (**d**) TGO.

**Figure 9 materials-12-02238-f009:**
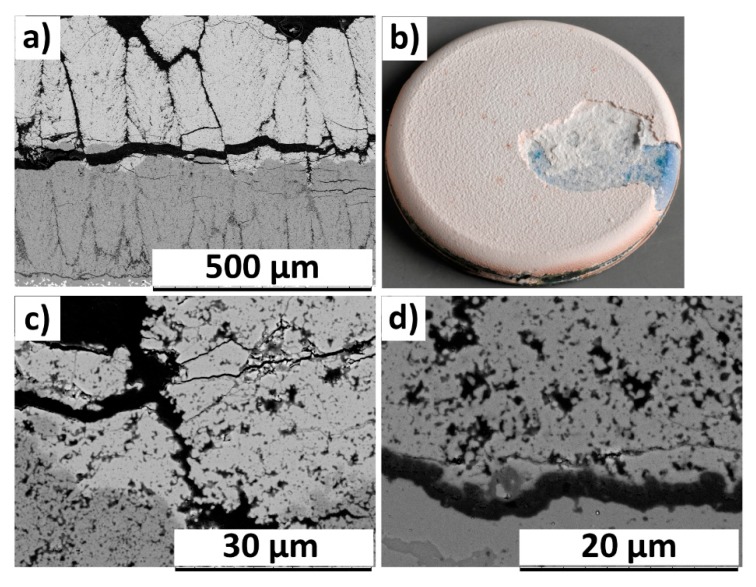
SEM micrograph of BRT failed (301 cycles) 250GZ/250YSZ (**a**) Cross section; (**b**) photograph; (**c**) high magnification cross sectional view; (**d**) TGO.

**Figure 10 materials-12-02238-f010:**
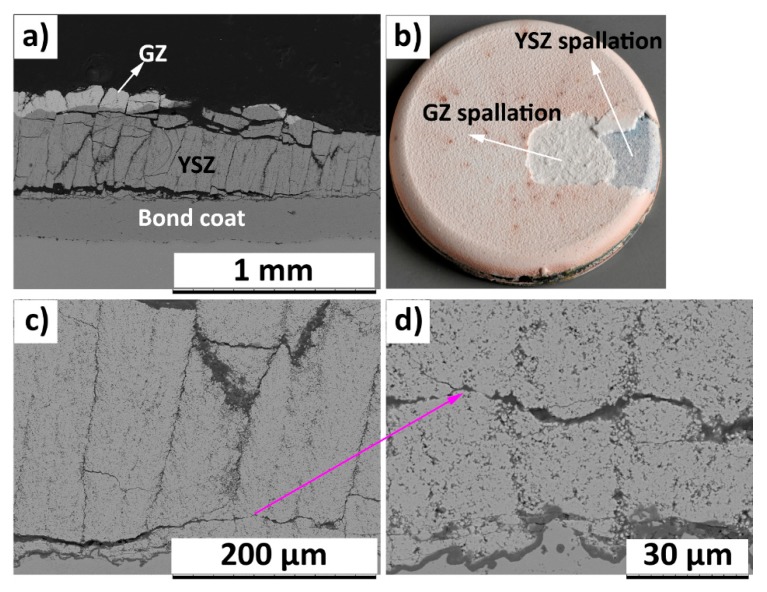
SEM micrograph of BRT failed (101 cycles) 100GZ/400YSZ (**a**) Cross section; (**b**) photograph; (**c**) cross sectional view at a different location; (**d**) TGO.

**Table 1 materials-12-02238-t001:** Suspension plasma spray parameters for YSZ and GZ layers in the three double layered TBC variants.

Parameters	YSZ	GZ
Solid load content (wt. %)	25	25
Median particle size (nm)	550	550
Solvent	Ethanol	Ethanol
Stand off distance (mm)	100	100
Enthalpy (kJ/I)	12.5	12.5
Atomizing gas flow (L/min)	20	20

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
