# Peer review of "Durability of Gadolinium Zirconate/YSZ Double-Layered Thermal Barrier Coatings under Different Thermal Cyclic Test Conditions"

_materials, 2019, doi:10.3390/ma12142238_

Round 1

Reviewer 1 Report

Journal: Materials 

Decision: Major 

 This manuscript entitled “Durability of gadolinium zirconate/YSZ double layered thermal barrier coatings under different thermal cyclic test conditions” deals with the thermal durability evaluation and elaboration of failure mechanism of double layered TBC under different thermal cyclic test conditions. The suggested design concepts and results are interesting and meaningful to the researchers in thermal barrier coatings. However, a few points need to address for supporting and substantiating their statements. In this regard, this manuscript would be suitable for publication after major revisions. Detailed comments are as follows:

1. The test results which is contrary to the author’s hypothesis, especially the BRT test, is needed to be strengthened with logical explanation. 

2. The poorest lifetime of 100GZ/400YSZ, which is only 20% cycle of 400GZ/100YSZ, is unacceptable. YSZ has higher fracture toughness as well as the higher porosity in as-prepared state which results in the better strain tolerance (shown in figure 2). Moreover, the delamination after BRT test is occurred at the interface between YSZ and GZ in common, implying the same failure mechanism. The authors should provide the logical elaboration of such results. 

3. How many specimens were conducted to TCF and BRT tests, respectively? Please provide the information. The deviation is too broad, especially in BRT test, it is even larger than the average value of the cycle number to failure. 

4. Line 39: it is needed to introduce and strengthen the purpose of the double layered design and features. 

5. Line 54: the author is suggested to describe the advantages of SPS, compared to conventional APS TBC. 

6. Line 152, 153: the specimen information is wrong. Please correct it. 

7. Line 211, 220: the failure mechanism of SPS-TBC is elaborated, citing some studies (ref, 12, 26) about APS-TBC. It is not appropriate because they have different microstructure and failure mechanism. 

8. Line 260: the author explained the BRT test results with sintering behavior, but provided no evidence which can confirm the suggestion. Please provide the evidence or strengthening elaboration.

Reviewer 2 Report

a very good paper.  attachment lists specific questions and comments

Reviewer 3 Report

This manuscript appears to be yet another serial publication on the work that the authors have been doing since 2015. A more detailed and through manuscripts are already published by the authors on almost similar materials. The information published in this manuscript is neither novel nor informative.  The manuscript fails to add value to the knowledge of the readers. The authors keep referring to the previous publications for important details of the experiments and in the discussion.  I started of marking the manuscript in the attached pdf until I figured out that this is just a compilation of some data that has been either published earlier or the data has not been analyzed in depth.  I also noticed that several versions of the similar study are published in variety of journals which is acceptable as long as the information adds value to the readers of a given journal. This particular manuscript fails to do that. The conclusions are not supported by results and the discussion revolves around parameters that are neither measured nor reported.

Round 2

Reviewer 3 Report

The revision is acceptable.